# Analysis of Runoff Trends and Drivers in the Haihe River Basin, China

**DOI:** 10.3390/ijerph17051577

**Published:** 2020-02-29

**Authors:** Huashan Xu, Yufen Ren, Hua Zheng, Zhiyun Ouyang, Bo Jiang

**Affiliations:** 1Beijing Urban Ecosystem Research Station, State Key Laboratory of Urban and Regional Ecology, Research Center for Eco-Environmental Sciences, Chinese Academy of Sciences, Beijing 100085, China; xuhuashan@mail.bnu.edu.cn (H.X.);; 2Changjiang Water Resources Protection Institute, Changjiang Water Resources Commission of the Ministry of Water Resources, Wuhan 430051, China

**Keywords:** Haihe River Basin, runoff, abrupt change, human activity

## Abstract

During the past decades, runoff has been highly influenced by climate change and human activities in Haihe River basin, and it is important to analyze the runoff trends and the drivers of its change to guide water resources management. The Mann–Kendall method and Pettitt test were conducted to analyze the hydrological and climate trends. Data from six sub-basins were used, including runoff at six representative hydrological stations and precipitation and air temperature at 49 meteorological stations. We used multiple-regression analysis and policy review to explore the influence of climate change and human activities on the runoff change at six sub-basins. According to the results, annual runoff showed a significant downward trend at six hydrological stations (*p* < 0.05), and the most probable change points at all stations showed up during the period from the late 1970s to the early 1980s. Moreover, the middle and late 1990s could be another probable abrupt change point at Luan River and Chaobai River. The declining trend of the annual mean precipitation at the six sub-basins was insignificant (*p* > 0.05), and there were no significant abrupt change points except the Zhang River area (*p* < 0.05). Compared with the precipitation trend, the annual mean air temperature exhibited a significant increasing trend at all stations, and the period from the late 1980s to the early 1990s might be the most probable abrupt change points at all four sub-basins. The trend analysis and the abrupt change point analysis suggest that mean air temperature is the main climate factor that will lead to the decline in the runoff time-series, while the insignificant downward trend of the precipitation might accelerate the downward trend of the runoff data. Through elevant policy measures, including land-use reform and the construction of the Three-North (north, northeast, and northwest China) Shelter Forest, China started to implement a family-contract responsibility system and initiated the first stage of construction of the Three-North Shelter Forest Program in 1978. The land-use reform policies greatly stimulated the peasants’ initiative for land management and significantly changed the land use pattern and water use quantity in the Haihe River basin in a short time. Besides, the precipitation decreased and the air temperature rose, so an abrupt change in runoff occurred from the late 1970s to the early 1980s. The abrupt change in the runoff in the middle and late 1990s highly tallied with the construction time of the Three-North Shelter Forest Program. After near 20 years of construction of the Three-North Shelter Forest Program, the forest area increased, the forest quality had been improved, and the vegetation coverage on the underlying surface had been changed significantly, so the construction of the Three-North Shelter Forest Program was an important cause of runoff change in the middle and late 1990s. Also, change in precipitation and air temperature enlarged the effect of change in the runoff.

## 1. Introduction

The Haihe River Basin is located in a semi-humid and semi-arid region, with a dense population and relatively high urbanization. It is also an important industrial and high-tech base in China. The North China Plain, which is partly located in the Haihe River Basin, is one of the three major grain production areas in China, forming an important strategic position regarding national economic development. The Haihe River Basin also experiences conflict between supply and demand for water resources, resulting in serious water environment deterioration and degradation. The ecological environment in the Haihe River Basin is in relatively poor condition. Excessive exploitation of water resources has resulted in river dry-up and enlarged groundwater funnels, with wetland areas also decreasing from 10,000 km^2^ in the 1950s to 1000 km^2^ at the end of the twentieth century [1]. Furthermore, water resource shortages and resulting environmental problems have seriously influenced social and economic development. Therefore, studies on hydrological changes and their driving factors in the Haihe River Basin are of importance for the optimal utilization and improved management of water resources, as well as supporting sustainable future social and economic development.

The Mann–Kendall (M–K) non-parametric statistical method [2] can be used to test variables with non-normal distribution, such as runoff trend changes [2,3,4,5,6]. The identification of change points in a hydrological data series is effective for studying the influences of climate change and human activities on hydrological and water resources. The Pettitt change point test is based on the non-parametric detection of change points within a data series [7] and uses a simple calculation procedure to identify change times and abrupt shifts in series distribution [8,9,10].

Previous studies have investigated runoff change trends and abrupt shifts in the Haihe River Basin and its sub-basins, including the whole Haihe River Basin [3,11,12]; Luan River [5,13,14,15]; Chaobai River [5,15,16]; Yongding River [15,17]; Daqing River [15]; Ziya River [15]; Zhangwei River [5,15]; Heihe River, a tributary of Chaobai River [16]; tributaries of Yongding River, including Yanghe River [18] and Sanggan River [19]; tributaries of Ziya River, including Hutuo River [5,20] and Yehe River [21]; and tributaries of Daqing River, including Juma River [15,22], Tanghe River [15,22], Shahe River [22], South Juma River [22], and Baigouyin River [22]. The runoff trend change results indicated descending trends at different significance levels. The most common period during which an abrupt shift in runoff occurred was from the late 1970s to the early 1980s, with several of the individual studies above revealing two change points in some sub-basins [13]. 

Although considerable research efforts have been made on runoff changes and driving forces in the Haihe River Basin, several questions remain unanswered. For example, how many change points of the upstream mountainous runoff have occurred in the six sub-basins of the Haihe River Basin over the past 50 years? Are the abrupt change situations consistent between sub-basins? Are the main causes of abrupt change related to natural factors or human activities? In this paper, the runoff change trends and abrupt change times upstream of the six sub-basins of the Haihe River Basin were studied from two aspects, that is, climate change and human activity. This study provides a scientific basis for the optimal management of water and land resources in the Haihe River Basin and similar regions of China.

## 2. Material and Methods

### 2.1. Study Region

The Haihe River Basin is located in northeastern China and is an important heavy industry and high-tech industrial base, as well as an area of substantial grain production. It contains 355 counties and districts, including the cities of Beijing and Tianjin, most of Hebei Province, the eastern and northern parts of Shanxi Province, northern parts of Shandong and Henan provinces, and small parts of the Inner Mongolia Autonomous Region and Liaoning Province. It covers a total area of 320,000 km^2^, including 186,900 km^2^ of mountains (58.7%) and 131,400 km^2^ of plains (41.3% of land area) (Figure 1).

The Haihe River Basin contains numerous rivers, with dispersed river systems, including the Haihe, Luanhe, and Tuhai-Majia river systems. The Haihe River system is divided into the North Haihe River system, which includes Jiyun River, Chaobai River, Beiyun River, and Yongding River, and the South Haihe River system, which includes Daqing River, Ziya River, Zhangweinan Canal, Heilonggang Yundong, and the Haihe River trunk stream. The Haihe River system has a basin area of 235,100 km^2^, and flows into the Bohai Sea at Tianjin City. The Luanhe River system has a basin area of 45,900 km^2^, and the Tuhai-Majia river system, which includes Tuhai River and Majia River, has a basin area of 33,000 km^2^, with both flowing into the Bohai Sea separately (Figure 1). The construction of the Three North Shelterbelt is a major forestry ecological construction project in areas where wind and sand hazards and soil erosion are very serious in China. The project planning starts from 1978 to 2050. The construction of the Yanshan Mountain Shelterbelt system in the Haihe River Basin was implemented in 1978. The Luanhe River, Chaobai River, Yongding River, and Daqing River Basin in this study area are all within the scope of the project. The project construction focuses on maintaining water and soil and conserving water sources. The completed afforestation area is 1.397 million hm^2^ [23]. The Beijing–Tianjin sandstorm source control project was launched in June 2000 and was fully launched in March 2002. The project construction period is from 2001 to 2010. The Luan River, Chaobai River, Yongding River, and Daqing River Basin are also in the scope of the project implementation [24] (Figure 1).

### 2.2. Data Source

Hydrological stations included Luanxian and Panjiakou reservoir stations (Luan River catchment), Sandaoying and Zhangjiafen stations (Chaobai River catchment), Xiangshuipu and Guanting reservoir stations (Yongding River catchment), Fuping and Xidayang reservoir stations (Daqing River catchment), Pingshan station (Ziya River catchment), and Guantai station (Zhang River catchment) (Figure 1). Runoff data were obtained from the Hydrological Year Books (1961–2010) published by the Hydrological Bureau of the Ministry of Water Resources of the People’s Republic of China. Meteorological data, including precipitation and temperature, covering 1961 to 2010 were obtained from the China Meteorological Administration.

### 2.3. Methods

#### 2.3.1. Mann–Kendall (M-K) Non–Parametric Test

The Mann–Kendall trend test [25], commonly known as the Kendall tau statistic, is a non-parametric test used to assess the significance of monotonic trends of hydro-meteorological variables [12]. Statistic (S) is first defined, followed by the calculation of S distribution variance. Standardization processing and continuous correction of S are then conducted to obtain the statistical trend test assessment value (Z). Z > 0 indicates an ascending trend and Z < 0 indicates a descending trend. If there is a changing trend, the magnitude is calculated using Sen’s slope estimation [4,12].

#### 2.3.2. Pettitt Abrupt Change Point Test

Based on the Mann–Whitney statistical function *U_t, N_*, and assuming that samples *x_1_*,…*x_t_* and *x_t+1_*,…*x_N_* are from one series, the Pettitt abrupt change point test is performed by calculating the number of times that the previous sample series exceeds the posterior sample series; if the number of times is zero, it is assumed that there is no abrupt change point in the sample series. The statistic *k*(*t*) is the maximum value of $\left|U_{t,N}\right|$ at the most significant abrupt change point *t*. For the specific calculation formula and relevant probability (*P*) significance test, see Liu et al. [4]. 

#### 2.3.3. Regression Analysis

Regression analysis was used to identify the main driving factors of runoff change in the Haihe River Basin [12]. On the basis of the sequential Pettitt test results and abrupt change time, the runoff data were divided into two or three periods. The relationship between precipitation and runoff for different periods was then analyzed and compared. Comparisons of correlations between runoff and precipitation for the two periods provided a good measure for the effect of human activity on runoff.

## 3. Results and Discussion

Climate change and human activities are considered to be two major factors driving changes in the hydrological cycle [1,26,27,28,29,30]. Climate change includes precipitation and potential evapotranspiration changes caused by temperature changes [15]; human activities include direct and indirect human activities, such as land use/cover change (LUCC) affects canopy interception, soil infiltration, land-surface evapotranspiration (ET) [31,32], and other hydrological parameters during rainfall, which in turn affect the hydrological regimes and runoff mechanisms of river basins [33,34]. Bare land, agricultural land, and settlements have lower evapotranspiration, and the greater the presence of vegetation and forest areas, the greater the evapotranspiration [31,34]. The relationship between runoff change, climate change, land cover change, and human activities is shown in Figure 2.

### 3.1. M–K Trend Test and Abrupt Change Detection for Precipitation and Air Temperature

#### 3.1.1. Air Temperature

The M–K and Pettitt abrupt change results of air temperature in the upstream areas of the six sub-basins are shown in Table 1 and Figure 3. From 1961 to 2010, the air temperature upstream increased significantly in all six sub-basins and exhibited an abrupt change from the late 1980s to the early 1990s. The time of abrupt change for Chaobai River, Daqing River, and Luan River was in 1988, 1993, and 1987, respectively, and for Yongdian River, Zhang River, and Ziya River was in 1993. These results are consistent with those obtained by Bao et al. [1] and Zheng et al. [35]; however, Liu et al. [22] found that the abrupt change point of air temperature in the Daqing River Basin occurred in 1981, approximately 10 years earlier than that found in our research. From 1961 to 2010, the change in air temperature in the six sub-basin-upstream areas of the Haihe River ranged from 0.019 °C/year to 0.033 °C/year, smaller than that reported previously for the whole Haihe River Basin (0.035 °C/year) [3]. This increase in air temperature has led to enhanced evapotranspiration and reduced runoff yield under equivalent precipitation conditions. Under the condition that the land cover is not changed, an increase in temperature will lead to an increase in evaporation, an increase in the transpiration of vegetation, and a decrease in the runoff coefficient. Under the same rainfall, the runoff will inevitably be reduced.

#### 3.1.2. Precipitation

The M–K test showed that precipitation exhibited a non-significant declining trend in the upstream areas of the six sub-basins of the Haihe River Basin from 1961 to 2010. The decrease in magnitudes ranged from 0.38 mm/year to 2.21 mm/year, relatively close to the 2.096 mm/year result obtained by Cong et al. [3]. Only the precipitation change in the upstream area of the Zhang River reached significance (0.05, Table 1). Wang et al. [5] also demonstrated that precipitation exhibited a decreasing trend in the Luan River, Hutuo River (a tributary of Ziya River), and Zhang River basins; however, only precipitation in the Hutuo River Basin reached a confidence level of 95%, with the changing trend not significant in the other two basins. Liu et al. [22] found that the abrupt change point of precipitation decrease in the Daqing River Basin occurred in 1981, whereas Bao et al. [1] found that the abrupt change in the Haihe River Basin occurred in 1979 (1951–2007) owing to a decrease in summer precipitation. Generally, precipitation has decreased in the Haihe River Basin in the last 50 years and is an important cause of runoff decrease in all sub-basins. However, the Pettitt test showed that precipitation change did not reach a significant level in any of the sub-basins of the upstream areas of the Haihe River Basin (Figure 4).

### 3.2. M–K Trend Test and Abrupt Change Detection for Runoff

The M–K trend test results of runoff at the 10 hydrological stations in the six sub-basins are shown in Table 2. Annual runoff exhibited significant decreases in all six sub-basins. The Pettitt detection curves in the six sub-basins displayed three different shapes, namely, “W” shaped, for example, Luan River and Chaobai River; “V” shaped, for example, Ziya River and Zhang River; and an indeterminate shape somewhere between a “W” and “V”, for exmaple, Yongding River and Daqing River (Figure 5).

As seen in Table 2 and Figure 4, an abrupt change in runoff in the Haihe River Basin occurred in the late 1970s to early 1980s (1977–1984), with an abrupt change also occurring in the mid to late 1990s (1996–1999) in Luan, Chaobai, Yongding, and Daqing rivers. The first abrupt change times obtained here are consistent with the results of other researchers, with relatively small differences in a year. These slight differences are likely because of differences in time series length, diverse methods for abrupt change analysis, and inconsistent significance levels selected.

### 3.3. Cause Analysis for Runoff Change

Climate change and human activities are two major factors that drive changes in the hydrological cycle [1,36]. Climate change includes changes in precipitation and air temperature caused by potential evapotranspiration fluctuations [15]. Direct human activities, including a rapid increase in population, industry, and cultivation areas; reservoir dam construction [37]; and diversion and extraction of surface and groundwater resources, can significantly change runoff in a basin [38]. Indirect human activities on runoff can occur after relevant soil and water conservation projects, such as tree planting and afforestation, are implemented. Under such projects, land use/cover can change, leading to increases in vegetation coverage, canopy interception, soil adjustment effects, and soil moisture capacity, as well as lengthening the runoff yield process, increasing evapotranspiration, and finally decreasing direct runoff [1,39].

#### 3.3.1. Relationship between Precipitation and Runoff

To analyse the relationship between precipitation and runoff, six hydrological stations (Luanxian, Zhangjiafen, Guanting Reservoir, Xidayang Reservoir, Pingshan, and Guantai) from the Luan River, Chaobai River, Yongding River, Daqing River, Ziya River, and Zhang River sub-basins were selected. Precipitation and runoff data from 1961 to 2010 were used to plot the precipitation–runoff relationship according to the abrupt change time, and linear analysis was conducted. As seen in Figure 6, the precipitation and runoff lines before the abrupt change are above those after the abrupt change, indicating that under equivalent precipitation conditions, the runoff generated before was larger than that after the abrupt change [1,5]. Thus, the relationship between precipitation and runoff before the abrupt change point was more significant than that after [12], suggesting that human activity was the main driving factor causing a runoff decrease in the basin. The underlying surface of the basin has not changed, and the river runoff is only affected by precipitation (except for extreme precipitation events). The precipitation–runoff relationship curve will only shift up and down, and the slope will remain basically unchanged. If the slope changes, the precipitation–runoff curve will deflect, indicating that the underlying surface of the watershed has changed.

#### 3.3.2. Land-Use Change

Changes in land use can significantly impact and modify runoff [40]. Forest land has the most significant effect on runoff, followed by cultivated land, grassland, residential land, and industrial and mining land, with water areas and unutilized land exhibiting insignificant effects [35]. When the cultivated area exceeds 25% in a sub-basin, abrupt changes in runoff can occur. For the Haihe River Basin, a larger ratio of arable land resulted in an earlier time of abrupt change in runoff and a more evident runoff decrease [12]. The family-contract responsibility system and the Beijing–Tianjin Wind and Sand Source Control Project have been the main driving forces for land-use change and transfer in the Luan River, Chaobai River, Yongding River, and Daqing River basins [35]. Since China implemented the family-contract responsibility system in 1978 to improve farmer initiative for land management, water appropriation and use from rivers has increased, resulting in a decrease in runoff [12]. Thus, an abrupt change in runoff occurred from the late 1970s to the early 1980s, with the resulting effects amplified by precipitation reduction.

The abrupt change in the Luan River, Chaobai River, Yongding River, and Daqing River sub-basins at the end of the last century (1996–1999) is a significant reflection of the effects of the Three-North Shelter Forest Program in China. This abrupt change was not found in the Ziya River or Zhang River sub-basins, which were not covered by the aforementioned program (Figure 1). The Three-North Shelter Forest Program was established in China in 1978 and is the largest environmental protection and ecological recovery project implemented in China. Since its establishment, the project has resulted in changed land-use patterns and increased vegetation coverage. In Luan River and Chaohe River, forest and grasslands were converted into cultivated lands before 1985, but cultivated lands were converted into forest and grasslands after 1990 owing to the implementation of the project [5]. In the Chaobai River sub-basin, the forest land area ratio increased from 48% in 1980 to 65% in 1995, the grass land area ratio decreased from 28% to 16%, and the cultivated land area ratio decreased from 22% to 17% [41]. Under equivalent precipitation conditions, forest vegetation increased and surface runoff decreased significantly. In the 1990s, the air temperature in the Haihe River Basin increased evidently, evapotranspiration was enhanced, vegetation coverage increased, and plant transpiration was strengthened, which hastened runoff decrease in the basin. During the late stage of the Three-North Shelter Forest Program and Beijing–Tianjin Wind and Sand Source Control Project, the forest area in the Haihe River Basin increased by 3.4% from 2000 to 2010, with the possibility that river runoff has continued to decrease since 2010 under equivalent precipitation conditions.

#### 3.3.3. Soil and Water Conservation Projects

To control soil erosion, soil-retaining dams were constructed in almost all valleys of the Haihe River Basin from 1958 to 1975 [5]. The construction of soil-retaining dams and re-shaping of slopes into terraces in the Chaobai River sub-basin in the early 1980s [41] (Figure 7) have markedly influenced runoff in the area [42]. Similarly, the upstream area in the Yongding River sub-basin has experienced serious soil and water loss and was listed as one of eight key control areas for soil and water conservation in China in 1983, with soil and water conservation projects implemented since the 1980s [17]. The implementation of these projects has effectively lessened soil erosion, but also significantly changed the runoff yield process and reduced runoff.

#### 3.3.4. Social and Economic Development

Increases in water use owing to social and economic development have resulted in considerable changes in the runoff in the Haihe River Basin. In the Luan River sub-basin, Xu et al. [2] stated that the main cause of the abrupt change in runoff at the Panjiakou Reservoir hydrological station in 1980 was increased water use owing to increased demands for domestic, industrial, and agricultural water. At Miyun Reservoir in the Chaobai River sub-basin, reservoir inflow declined from 90.3 mm/a before 1984 to 41.8 mm/a and after 1984, yet water use increased from 2.2 mm/a to 13.4 mm/a (11.2/48.5) [43]. In the Zhang River sub-basin, to meet irrigation water demands (irrigation district area of up to00 km^2^), three relatively large diversion projects, that is, Yuefeng Channel (1975), Yuejin Channel (1977), and Hongqi Channel (1969), were constructed in succession, and account for a water diversion flow rate of 93 m^3^/s [6]. Thus, the increase in water demand due to social and economic development has significantly changed river runoff in the Haihe River Basin.

The decrease in runoff observed in the Haihe River Basin is the result of the joint effects of climate change and human activities. Intense human activities have had a drastic influence on the hydrological processes in the basin, with water use increase, land-use change, construction of major ecological projects, implementation of soil and water conservation projects, and social and economic development all influencing runoff. Under the background of climate change, optimizing land use is an effective way to achieve scientific management of water resources in the Haihe River Basin [44]. As the South-North Water Transfer Project, which was implemented in 2012, exerts beneficial results, the shortages of water resources and relevant environment problems in the Haihe River Basin will be alleviated to a certain extent. However, reasonable land use, moderate soil and water conservation, strict water resource management, and good water use habits are the most effective long-term means to alleviate water resource shortages in the Haihe River Basin.

## 4. Conclusions

(1) Precipitation in the six sub-basins of the Haihe River Basin has exhibited a decreasing trend, but only reached a significant level upstream of the Zhang River, and no abrupt change occurred in any sub-basin. Air temperature increased significantly, with an abrupt change occurring from the late 1980s to the early 1990s. Precipitation decrease and air temperature increase have enlarged the runoff change effect in the Haihe River Basin.

(2) Basin-wide abrupt change in runoff occurred upstream of all six sub-basins of the Haihe River Basin from the late 1970s to early 1980s. The main cause of this abrupt change in runoff was land reform policies implemented in China in 1978, which stimulated farmer initiatives for land management, resulting in an increase in water use.

(3) Abrupt changes in runoff occurred upstream of the Luan River, Chaobai River, Yongding River, and Daqing River sub-basins by the end of the 1990s (1996–1999). These four sub-basins were entirely (Luan River, Chaobai River, and Yongding River sub-basins) or partially (Daqing River sub-basin) located within the implementation range of the Three-North Shelter Forest Program in China, which caused forest vegetation change, resulting in river runoff decrease in the basin.

(4) Changes in land surface cover caused by the implementation of major ecological projects, and further quantification of their impacts on runoff changes, will become the focus of future research.

## Figures and Tables

**Figure 1 ijerph-17-01577-f001:**
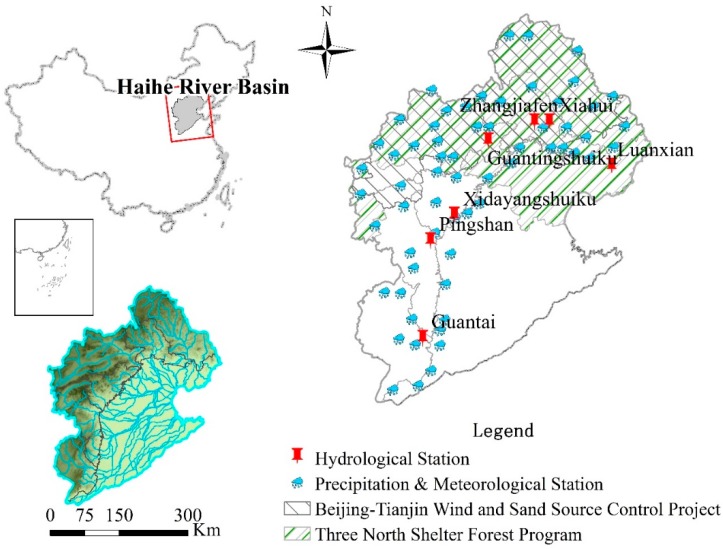
Areas encompassing the implementation of the Three North Shelter Forest Program and Beijing-Tianjin Wind and Sand Source Control Project in the Haihe River Basin, as well as the distribution of hydrological stations.

**Figure 2 ijerph-17-01577-f002:**
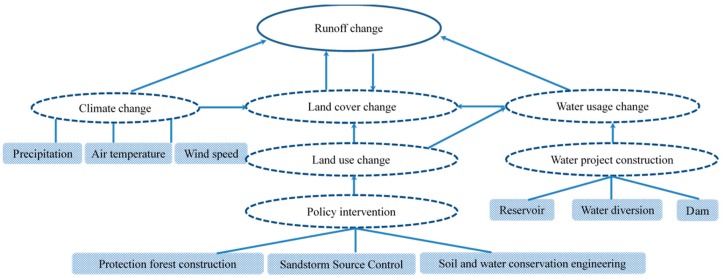
Impact of climate change and human activities on runoff changes.

**Figure 3 ijerph-17-01577-f003:**
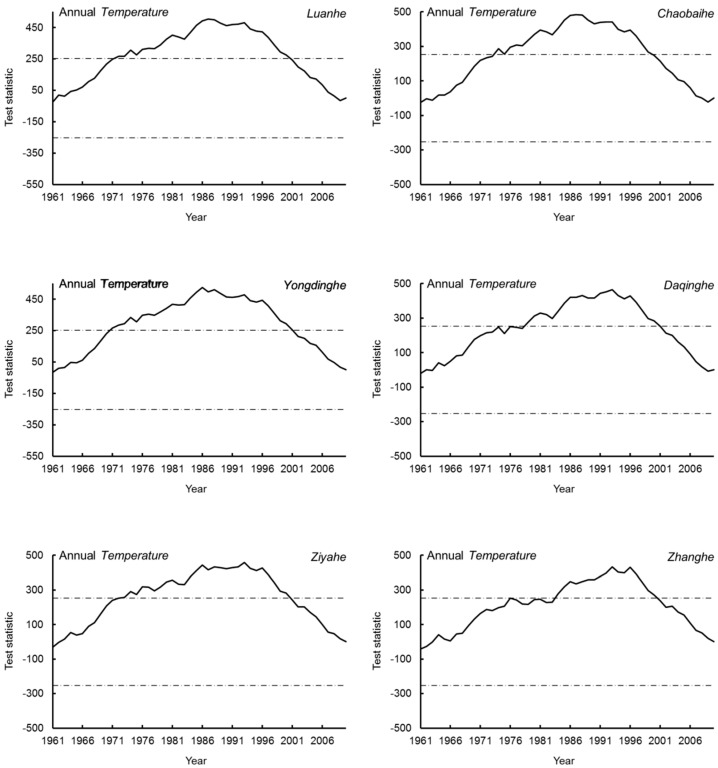
Detection results of the abrupt change point of air temperature in the six sub-basins of the Haihe River Basin.

**Figure 4 ijerph-17-01577-f004:**
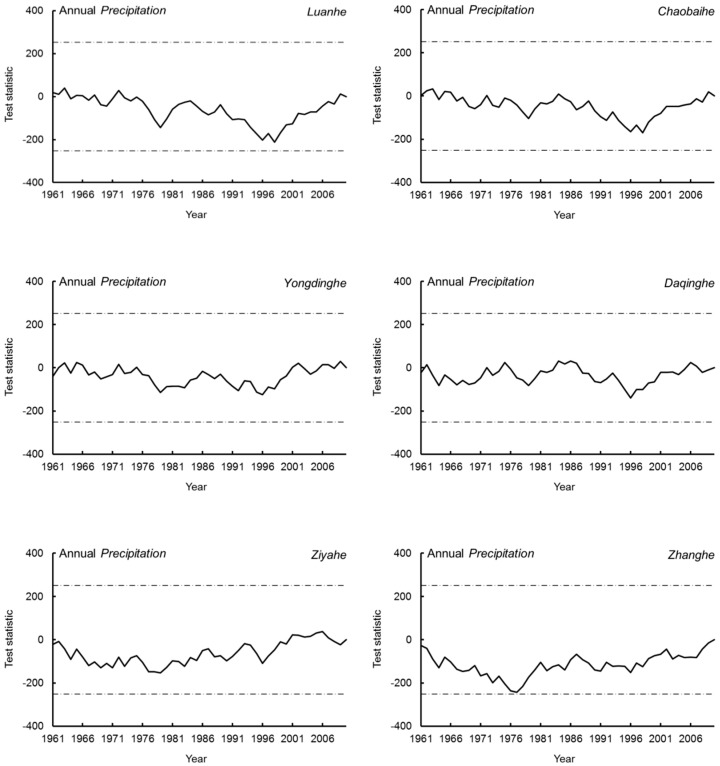
Detection results of the abrupt change points of precipitation in the six sub-basins of the Haihe River Basin.

**Figure 5 ijerph-17-01577-f005:**
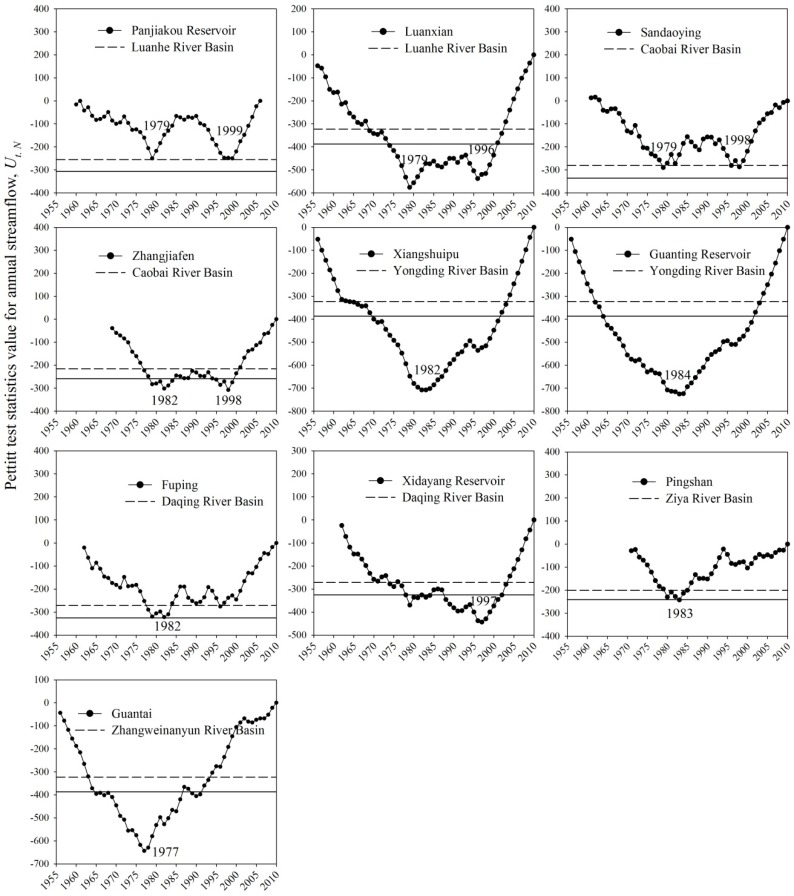
Abrupt change detection results of runoff in the six sub-basins of the Haihe River Basin.

**Figure 6 ijerph-17-01577-f006:**
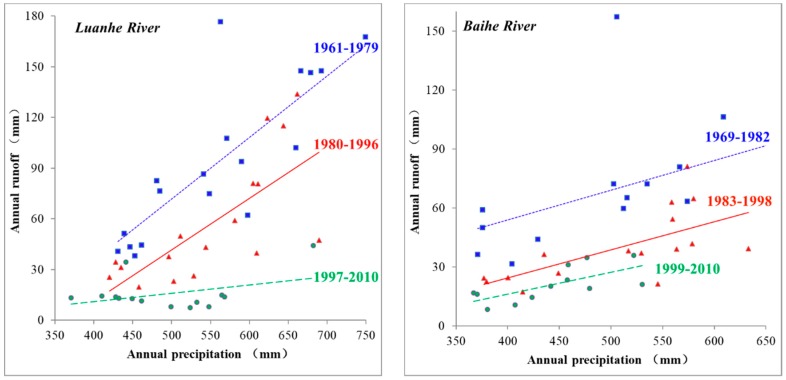
Relationship between precipitation and runoff in the upstream areas of the Luan River, Yongding River, Ziya River, and Zhang River sub-basins.

**Figure 7 ijerph-17-01577-f007:**
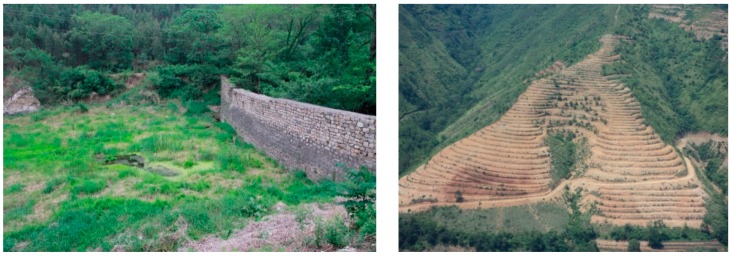
Soil-retaining dam and slopes reshaped into terraces in the Haihe River Basin.

**Table 1 ijerph-17-01577-t001:** Mann–Kendall (M–K) trends of precipitation and air temperature in upstream mountainous areas of the six sub-basins of the Haihe River Basin.

Sub-Basin	Time Series	Stations	Precipitation		Temperature	
			Z	Sen’s Slope	Z	Sen’s Slope
Luan River	1961–2010	12	−0.89	−0.78	4.60 **	0.031
Chaobai River	1961–2010	6	−0.69	−0.59	4.18 **	0.027
Yongding River	1961–2010	18	−0.52	−0.38	4.77 **	0.033
Daqing River	1961–2010	9	−0.64	−1.07	4.02 **	0.022
Ziya River	1961–2010	7	−0.97	−1.26	4.40 **	0.025
Zhang River	1961–2010	5	−2.02 *	−2.21	3.48 **	0.019

* and ** represent significance at the 0.05 and 0.01 levels, respectively.

**Table 2 ijerph-17-01577-t002:** Trend changes in annual runoff and detection results of abrupt change in runoff in the six sub-basins of the Haihe River Basin.

Hydrological Station	Time Series	Sub-Basin	Trend Change		Possible Abrupt Change Time	
			Z	Sen’s Slope	First Abrupt Change Year	Second Abrupt Change Year
Panjiakou Reservoir	1960–2006	Luan River	−2.56 *	−0.644	1979	1999
Luanxian	1956–2010	Luan River	−5.19 **	−1.663	1979	1996
Sandaoying	1961–2010	Chaobai River	−2.27 *	−0.821	1979	1998
Zhangjiafen	1969–2010	Chaobai River	−4.18 **	−1.103	1982	1998
Xiangshuibao	1956–2010	Yongding River	−7.22 **	−0.705	1982	1996
Guanting Reservoir	1956–2010	Yongding River	−7.70 **	−0.592	1984	1997
Fuping	1962–2010	Daqing River	−3.18 **	−1.728	1982	1996
Xidayang Reservoir	1962–2010	Daqing River	−5.22 **	−2.260	1979	1997
Pingshan	1962–2010	Ziya River	−3.61 **	−1.169	1983	-
Guantai	1956–2010	Zhang River	−4.59 **	−1.319	1977	-

* and ** represent significance at the 0.05 and 0.01 levels, respectively.

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
