# Peer review of "Analysis of Runoff Trends and Drivers in the Haihe River Basin, China"

_ijerph, 2020, doi:10.3390/ijerph17051577_

Round 1

Reviewer 1 Report

The authors dealt with the hydrological changes in the Haihe River basin by a few statistical techniques. Especially, they tried to detect the gradual change (or trend) and the abrupt change by MK trend test and Pettitt's change point test. Also, the authors try to reveal the relationship between runoff and other variables such as precipitation and air temperature. I think that the selected theme is very important and also, relevant to International Journal of Environmental Research and Public Health. However, I think that this article should be revised before publication. Please, see the attached pdf file for the detailed comments from me.   

Author Response

The focus of this article is to test the characteristics of long-term serial runoff changes using traditional and mature statistical methods, and to analyze whether the runoff abrupt time coincides with temperature and precipitation, and related national policies and administrative intervention measures at the time node, and also to find out the corresponding change characteristics. All these are to provide research direction and research focus for later quantitative modeling, so the innovation of theory and methods will not be the focus of this article. In the next step, we will analyze land use changes, build evapotranspiration models based on vegetation changes, temperature and other relevant factors, and perform quantitative demonstration. The results will also be presented in the form of dissertations.

For the Detailed,Please see the attachment.

Reviewer 2 Report

 On account of the manuscript IJERPH-692232, entitled “Analysis of runoff trends and drivers in the Haihe River Basin, China” by Xu Huashan et al., the authors investigated the runoff change trends, precipitation, and air temperature data from the upstream mountainous areas of the Haihe River Basin as an important industrial and hi-tech base in China, and their trends and change point times were tested using the Mann-Kendall and Pettitt methods. The topic is important to conduct water resource management in the river basin in China. After careful consideration, I feel that this manuscript is to be published after improvement of some minor shortcomings. Details of my comments are as follows:

 The manuscript was well written and designed, and the authors got interesting results. Several revisions are, however, required before publication. P.3, Line 83-84; description of “with both flowing into the Bohai Sea separately (Error! Reference source not found.).” is to be corrected with the appropriate references, and the Reference list is also to be updated accordingly. After that I am ready to recommend the present manuscript for publication.

Reviewer 3 Report

The aim of the paper is evaluation of time trends and change point detection of runoff, precipitation, and air temperature in mountainous areas of Haihe River Basin. Mann-Kendall and Pettitt methods had been used. The paper should be improved significantly before publication in MDPI journal.

Please make the abstract to more communicative.  Follow the good practise of academic writing: provide research question – what’s scientific problem? Then describe the methodologic framework, draft the results and provide answer to Your question… If want to extend the abstract You can add some intentions (why Your research is important) and implications (what consequences are expected for the practice for example).

Fig. 1. Make the maps more readable, please. The concepts of the Three North Shelter Forest Program and Beijing-Tianjin Wind and Sand Source Control Project shall be described in the text body before visualisation on the maps. I suggest to change the point features symbology – please avoid the infantile symbols…

Line 83: Please prepare the manuscript more carefully: Error! Reference source not found.

2.2. Data source – as it states:  In this paper, the runoff change  trends and abrupt change times upstream of the six sub-basins of the Haihe River Basin were studied from two aspects, i.e., climate change and human activity; there is no data on human impact! Enhance the section significantly.

2.3 Methods – provide research framework or workflow graphic. How did You incorporate human factors into runoff study?

Results and Discussion. Please provide descriptive statistics for time series. The original data are not known for international readers.

Line 179: provide references

Line 194-195 suggesting that human activity was the main driving factor causing runoff decrease in the basin. Please discus that thesis more thoroughly… Provide evidence-based explanation.

Sections 3.3.2. and 3.3.3. – I strongly suggest rewriting the data source and methods subsections

Personally I’m not sure the conclusion no. 4 is based on Your own research, it is like opinion or expression of good political will – reconsider this paragraph please

Reviewer 4 Report

This paper must be improved before publishing in this journal.

General observation

In the introduction part author's must have references about this subject from all wourld not just from Asia region The methods part must be rewritten to descrie better the methods used (formula and results obtained before by the author's or others researches) Figure 2 and 3 must be replace în another type of graphics Figure 4 must be re-arraged. Table 3 must be deleted from this part and references about the researches must made in the introduction part. The period used, if its possible, must be update until 2016 or more. The references must be rewritten with more references around the wourld. 

Reviewer 5 Report

Dear Authors,

in the attached file there are my comments to your paper.

Kind regards.

Round 2

Reviewer 1 Report

The comments and questions from me have been revised and replied. So, I think that this article can be published as this present form.

Author Response

Reviewer 1 has no new comments.

Reviewer 2 Report

 On account of the manuscript IJERPH-692232R1, entitled “Analysis of runoff trends and drivers in the Haihe River Basin, China” by Xu Huashan et al., the authors the authors revised the manuscript appropriately according to the Reviewers comments. After careful consideration, I made a decision that the manuscript is acceptable for publication in its present form.

Author Response

Reviewer 2 has no new comments.

Reviewer 3 Report

The paper can be published in present form.

Author Response

Reviewer 3 has no new comments.

Reviewer 4 Report

The authirs have made general and specific modificați required. The paper can be published in this form

Author Response

Reviewer 4 has no new comments.

Reviewer 5 Report

The paper has been improved and the authors replied quite well to the reviewers' comments.

Author Response

Reviewer 5 has no new comments.